

# Accuracy of high-frequency musculoskeletal ultrasound in hand trauma: a retrospective surgical field-based validation study at a single center

Jiajia Wang[1,*], Lan Gao[1,*], Mingdi Fang[1], Fan Jiang[1] and Mei Peng[1]

Medical Ultrasound, Department of Ultrasound Medicine, the Second Affiliated Hospital of Anhui Medical University, Hefei, China
[*] These authors contributed equally to this work.

## ABSTRACT

**Background**. This study aims to assess the diagnostic accuracy of high-frequency musculoskeletal ultrasound in hand trauma patients, with a specific focus on tendon injury, avulsion fracture, and nerve damage, compared to surgical observations.
**Methods**. A retrospective study was carried out on 47 hand trauma patients who received high-frequency ultrasound examinations followed by surgical intervention from January 2022 to December 2024. Ultrasound assessment with the echoes, continuity, and alterations of tendons, nerves, and bone cortex, and the surgical findings were compared with the ultrasound features. The diagnostic accuracy of ultrasound was evaluated according to surgical outcomes.
**Results**. High-frequency ultrasound exhibited excellent diagnostic accuracy for tendon ruptures, achieving 92.86% for flexor tendons and 100% for extensor tendons, as well as 100% for avulsion fractures. Nonetheless, the accuracy was lower for tendon tears and nerve injuries, with 33.33% and 40%, respectively. Adding passive motion during ultrasound examination can enhance the accuracy of diagnosing tendon injury.
**Conclusion**. High-frequency ultrasound provided a dependable imaging technique for the diagnosis of tendon rupture and avulsion fracture in hand trauma. It may enhance clinical assessment and guide treatment selection for hand trauma patients.

## INTRODUCTION

Hand trauma is a common clinical condition frequently resulting from diverse mechanical injuries, including occupational accidents, falls, and sports activities, encompassing pathologies such as tendon injuries, nerve damage, and fractures (*Trybus et al., 2006*; *De Jong et al., 2014*). Given the intricate anatomy of the hand, delayed or inadequate assessment and treatment of injuries can result in dysfunction, significantly affecting the patient's quality of life (*Papes et al., 2023*).

Corresponding author
Mei Peng, 13955125956@163.com

The evaluation of hand trauma mostly relies on clinical examination, including the evaluation of swelling, flexion and extension function of the fingers, and sensory impairment. However, this method is inherently subjective, may not accurately determine the precise location and the extent of tendon injuries, impeding surgical treatment (*Habib et al., 2023*). While X-ray and CT are valuable for detecting hand fractures, they demonstrate limited sensitivity for small avulsion fractures and poor detection of tendon/nerve injuries (*Crombach et al., 2020*). Although MRI provides detailed tendon visualization, it fails to capture their dynamic movement, potentially resulting in false-negative findings. Furthermore, the lengthy wait times for MRI appointments is a crucial challenge in emergency, limiting its utility for acute hand injuries (*Verhagen & Chesaru, 2016*).

With the widespread use of high-frequency musculoskeletal ultrasound in clinical, the improvement of ultrasound resolution, and the sonographer's ability to recognize the anatomy of the hand, high-frequency musculoskeletal ultrasound has gradually been applied to the evaluation of hand disorders (*Chu et al., 2023*). Previous studies have mostly focused on the evaluation of carpal tunnel syndrome, and tenosynovitis (*Owers et al., 2007*). To our knowledge, a few of research using ultrasound to evaluate traumatic hand according to surgical findings as the gold standard. In this study, we retrospectively collected 47 hand trauma patients and described the ultrasound features and diagnostic accuracy, with the aim of providing descriptive insights into clinical assessment and surgical planning. The findings are presented as descriptive outcomes to inform practice, rather than to support statistical generalizations.

## MATERIALS & METHODS

### Patients

This is a retrospective observational study. All the research processes were performed by the ethical guidelines of the Declaration of Helsinki. This study received approval from the Ethics Committee of the Second Affiliated Hospital of Anhui Medical University (SL-YX2024-052).

We retrospectively and continuously enrolled hand trauma patients from January 2022 to December 2024 at the Second Affiliated Hospital of Anhui Medical University. All study participants provided written informed consent prior to data collection. Inclusion criteria: (1) the patients suffered hand trauma; (2) the patients underwent high-frequency musculoskeletal ultrasound examination of the hand; (3) the patients underwent surgical treatment to obtain a clear surgical field. Exclusion criteria: (1) the patients who were unable to receive comprehensive hand ultrasound examination due to the excessive scope of skin injury or serious area of infection; (2) the patients whose hands were in absolute passive position or who were unable to receive passive motion due to obvious pain during examination; (3) the patients who were unable to obtain clear ultrasound images due to the poor quality of ultrasound images.

Following the inclusion and exclusion of criteria, forty-seven patients were ultimately included in the study, as illustrated in Fig. 1.

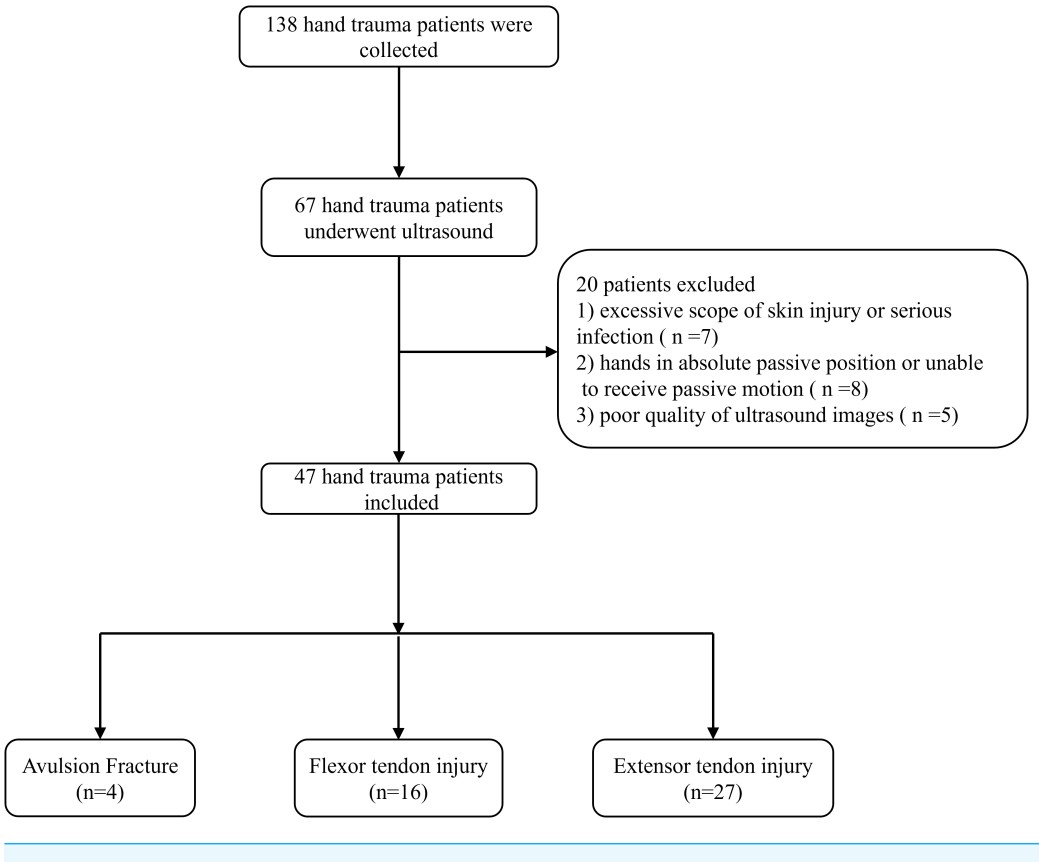

**Figure 1** **The flow chart of study.**

## The examination of ultrasound

Prior to the examination, inquired about the injury history, including the location, timing, and progression of the injury, and inquired the presence of clinical symptoms such as pain, swelling, and numbness. Request the patient to perform specific hand exercises including finger extension and flexion, wrist and flexion, to assess for hand dysfunction.

The Canon Aplio i800 and i900 color Doppler ultrasound systems were used, both equipped with i18LX5 and i24LX8 probes, with probe frequencies of 5–18 MHz and 8–24 MHz, respectively.

Patients were positioned comfortably to fully expose the hand trauma site for ultrasound examination. If skin lesions are present, it is essential to prioritize probe isolation and personal protection to minimize the risk of cross-infection.

The examination process starts with a point-to-point examination of the injured site, checking for fluid accumulation, hematoma, tissue swelling, or other abnormalities. Additionally, abnormalities in the bone cortex are noted, with a specific focus on the tendons and nerves in both longitudinal and transverse sonogram. Internal echo and continuity with tendon are evaluated, and if necessary, a passive motion examination is conducted for patients suspected of tendon injury. Tendon rupture patients need recorded the horizontal position of the rupture, morphology of the ruptured ends, and

distance between the ruptured ends. For tendon tears, the horizontal position, extent, and thickness of the tear are recorded. Diagnostic criteria followed standardized musculoskeletal ultrasound classifications: tendon rupture was defined as complete discontinuity of tendon fibers with retraction of torn ends; tendon tear as partial disruption involving >50% of tendon thickness; tendon swell as focal thickening >20% compared to adjacent normal tendon with decreased echogenicity; and tendon thin as focal thinning <50% of normal tendon thickness with loss of fibrillar patter (*Martinoli, 2010*).

All ultrasound examination are performed by experienced sonographers over 8 years of expertise in musculoskeletal ultrasound.

### Surgery

Following ultrasound exam, the patient undergoes a surgical treatment to carefully observed injuries of tendons, nerves, and bones. The appropriate surgical approach is taken according to the damage.

The diagnostic accuracy of preoperative musculoskeletal ultrasound was assessed using surgical field observation records as the reference standard.

### Statistical analysis

Measurement data were presented as mean ± standard deviation, while count data were expressed as N (%). The ultrasound diagnostic results were classified as true-positive (TP), true-negative (TN), false-positive (FP), and false-negative (FN) based on the diagnostic outcomes of the surgical field. The accuracy rate was computed using the formula: accuracy rate = (TP + TN)/(TP + TN + FP + FN) ×100%. Due to the limited sample size (47 cases), all analyses were restricted to descriptive accuracy rates presented as percentages with absolute counts. No inferential statistics were applied, and the results are intended for descriptive interpretation rather than statistical generalization.

## RESULTS

### Baseline characteristics of patients with hand injury

Forty-seven patients, comprising 32 males and 15 females with average age of (37.36 ± 16.52) years, were ultimately enrolled in the study. Among them, 28 patients sustained right hand injuries, while 19 patients sustained left hand injuries.

Five patients previously underwent hand trauma surgery. Among them, two patients exhibited thumb dyskinesia, one patient had limited wrist flexion, and one each showed dyskinesia in the index and ring fingers respectively. None of these patients reported swelling or numbness upon initial evaluation.

Of the remaining 42 patients without prior hand trauma surgery, five patients exhibited dyskinesia and numbness, six patients showed dyskinesia and swelling, and 31 patients presented with dyskinesia alone (15 patients had dysextension, 16 had dysflexion).

### Ultrasound features of hand trauma patients
#### *Avulsion fracture*

Four patients presented with avulsion fractures on ultrasound images. One patient exhibited intact extensor tendon (ET) continuity of the middle finger with normal morphology,

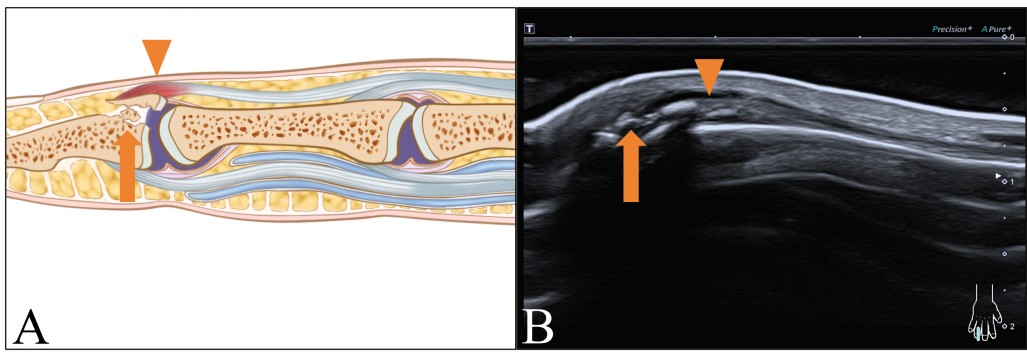

**Figure 2** **Avulsion fracture with ET injury in distal interphalangeal (DIP) joint.** (A) Schematic and (B) longitudinal sonogram. Arrow display free bone fragments in DIP, arrow head demonstrate swelled ET.

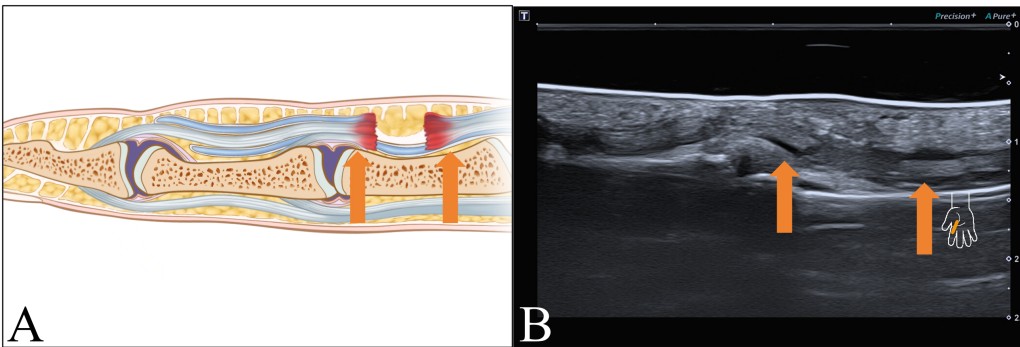

**Figure 3** **FT rupture.** (A) Schematic and (B) longitudinal sonogram. Arrow display tendon ends.

localized effusion, and free bone fragments at the distal phalanx (DP) attachment site, indicative of a single avulsion fracture. The remaining three patients displayed intact ET continuity of the little finger, localized tendon swelling, and free bone fragments at the DP attachment site, indicating an ET injury combined with an avulsion fracture, as illustrated in Fig. 2.

### Flexor tendon injury

Sixteen patients exhibited abnormal FT images: eight patients presented injuries both with the flexor digitorum profundus (FDP) and flexor digitorum superficialis (FDS), with seven cases of ruptures and one case of tear (Fig. 3).

Four patients had a rupture of the FDP and normal FDS. Additionally, two patients showed a rupture of the FDP alongside a tear of the FDS. Furthermore, one patient displayed a rupture of the FDS while the FDP appeared normal. One patient was re-examined after previous trauma surgery, with the ultrasound image illustrating a radial carpal flexor rupture along with a median nerve rupture.

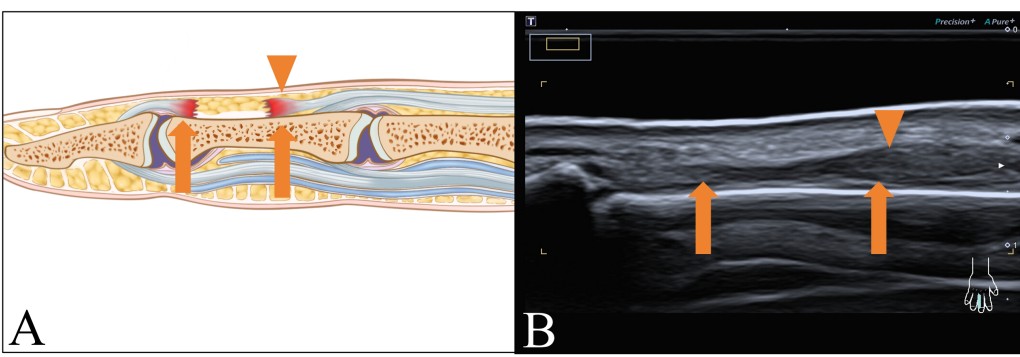

**Figure 4  ET rupture.** (A) Schematic and (B) longitudinal sonogram. Arrow display tendon ends, and arrow head demonstrate swelled tendon.

### ET injury

In 27 patients, 29 ETs ultrasound abnormalities were observed, unrelated to avulsion fractures.

Twenty-one patients experienced ET ruptures, as illustrated in Fig. 4. Two of them suffered from multi-tendon injuries, one had ET ruptures both in the index and middle fingers, and another had ET ruptures both in the middle and ring fingers. One of 21 patients had combined digital nerve echo inhomogeneity.

Six patients presented with tendon tears. Two patients exhibited partial disruption of muscle fiber continuity, three patients presented with localized ET swelling, while one patient showed localized ET thinning in ultrasound image.

### Passive motion examination in ultrasound

Static ultrasound failed to provide a definitive diagnosis in 29 patients, with 13 presenting flexor injuries and 16 presenting extensor injuries. Following a passive motion examination, tendon movement was absent in 24 patients, leading to a diagnosis of tendon rupture *via* ultrasound. Among the remaining five patients with observable tendon movement, two exhibited slight passive motion, resulting in a diagnosis of tendon tear upon ultrasound evaluation. The remaining three patients, who demonstrated good passive motion, were diagnosed with tendon injury through ultrasound assessment.

### Indirect ultrasound performance

Seven patients exhibited hematoma surrounding the injured tendons. Four patients displayed joint cavity effusions, one of which showed joint capsule bulging due to the effusion.

Detailed ultrasound performance recordings are shown in Table 1.

### Consistency of ultrasound results and surgical findings

The descriptive concordance between ultrasound findings and surgical exploration is summarized as follows:

**Table 1  Ultrasound performance of 47 patients with hand trauma.**

|  | All | Thumb | Index finger | Middle finger | Ring finger | Little finger |
|---|---|---|---|---|---|---|
| **Tendon (left/right)** | | | | | | |
| Left (number) | 19 | 6 | 3 | 2 | 3 | 5 |
| Right (number) | 30 | 8 | 6* | 6*# | 4# | 6 |
| **FT (number)** | | | | | | |
| rupture | 15 | 6 | 2 | 2 | 4 | 1 |
| tear | 1 | 1 | | | | |
| ET (number) (without avulsion fracture) | | | | | | |
| rupture | 23 | 6 | 4* | 3*# | 1# | 7 |
| tear | 2 | 1 | 1 | | | |
| swell | 3 | | 2 | 1 | | |
| thin | 1 | | | | 1 | |
| **Avulsion fracture (case)** | 4 | | | 1 | | 3 |
| **Indirect performance (case)** | | | | | | |
| joint cavity effusions | 3 | 2 | 1 | 1 | | |
| hematoma | 7 | | 2 | 3 | 2 | |
| joint capsule bulge | 1 | | | 1 | | |
| **Damage horizontal position (number)** | | | | | | |
| the wrist joint | 6 | 6 | | | | |
| metacarpal bones | 2 | | 2 | | | |
| phalanges | 41 | 7 | 6* | 9*# | 8# | 11 |
| **Passive motion (case)** | | | | | | |
| disappear | 24 | 8 | 4 | 2 | 5 | 5 |
| exist | 5 | 2 | 1 | 1 | 1 | |

Notes.
*One case had ET ruptures both in the index and middle fingers.
#One case had ET ruptures both in the middle and ring fingers.

Surgical exploration revealed that in 14 patients with FT rupture, the correct diagnosis by ultrasound after passive motion was 92.86% (13/14), and only one patient was misdiagnosed as a tear.

Six patients were surgically confirmed to have ET tears, with ultrasound correctly diagnosing 33.33% (2/6) of these cases. Four patients did not receive accurate ultrasound diagnoses, including one misdiagnosed as ET rupture and three incorrectly identified as ET only swelled.

Five patients had tendon injuries accompanied by localized nerve injuries, and a definitive ultrasound diagnosis was obtained only in two cases, with a correct ultrasound diagnosis rate of 40.00% (2/5).

**Table 2  Consistency between ultrasound and surgical findings in patients with hand trauma.**

| Surgery | | | Correctly diagnosed without passive motion (case) | Accuracy of ultrasound (%) | Correctly diagnosed after passive motion (case) | Accuracy after passive motion (%) |
|---|---|---|---|---|---|---|
| | Surgical findings | N | | | | |
| Tendon rupture | Rupture of FT | 14 | 3 | 21.43% | 13[*] | 92.86% |
| | Rupture of ET | 18 | 9 | 50.00% | 18 | 100% |
| Tendon tear | Tendon tear of FT | 0 | / | | | |
| | Tendon tear of ET | 6 | 1 | 16.67% | 2[#] | 33.33% |
| Tendon injury combined nerve injury | | 5 | 2 | 40.00% | 2 | 40.00% |
| Tendon injury combined avulsion fracture | | 4 | 4 | 100.00% | / | |
| All | | 47 | 19 | 40.42% | 35 | 74.47% |

Notes.
[*]One case of FT rupture was misdiagnosis of a tear since peripheral effusion resulting in a false positive for passive motion.
[#]Four cases of ET tear was misdiagnosis of rupture in ultrasound, due to the presence of hematoma around the tendon: passive motion negative in one case; no obvious ultrasound feature of tendon tear with good passive motion in three cases.

A definitive preoperative diagnosis was achieved in all four patients with combined avulsion fractures, yielding a 100% diagnostic concordance rate. Table 2 illustrates the concordance between ultrasound and surgical findings.

## DISCUSSION

In this study, we retrospectively analyzed ultrasound images from 47 hand trauma patients, using the surgical field of view as the gold standard to evaluate the accuracy of high-frequency musculoskeletal ultrasound. Our descriptive findings suggest that high-frequency ultrasound may support the assessment of FT and ET injuries in hand trauma, particularly through dynamic techniques like passive motion. This aligns with clinical needs for preoperative localization of tendon injuries, but the results should be interpreted as preliminary due to the sample size.

Comprehensive musculoskeletal ultrasound examinations have demonstrated the diagnostic accuracy of FT rupture and ET rupture could be up to 92.86% and 100%, respectively. In previous studies, hand trauma assessments depended heavily on the physician's empirical clinical examination, limited by ultrasound probe frequency and the spatial resolution of other image examination (*Sehmbi et al., 2023*). Previous studies have pointed out the limitations of clinical examination, such as the inability to accurately assess the horizontal position of tendon injuries before surgery, which makes surgical treatment difficult. In the current investigation, 21.28% (10/47) of patients exhibited joint effusion, localized hematoma, and joint capsule swelling, which were the factors that leading to false-negative results in clinical examination. Nevertheless, these performance observed in ultrasound images can serve as valuable indicators for ultrasonographers in diagnosing hand tendon injuries, thereby enhancing the sensitivity of ultrasound-based diagnoses (*Dugom et al., 2023*). While previously studies explored the MRI have potential in evaluating conditions like stenosing tenovaginitis, proliferative tenosynovitis, and other degenerative, metabolic, and systemic inflammatory wrist disorders (*Chen et al., 2023*).

However, the limitations of MRI in diagnosing traumatic tendon ruptures may stem from its inability to facilitate dynamic examination of the patient, a capability in which ultrasound demonstrates superior performance.

Out of the 47 patients, 43 presented with tendon injuries. Among them, 80.85% (38/47) had isolated tendon injuries, while 10.64% (5/47) had tendon injuries in conjunction with nerve damage. This finding underscores that tendon injuries are the predominant type of injury in patients with hand trauma (*Middleton, 2022*). In this study, 18 cases of ET rupture and 14 cases of FT rupture were confirmed by surgery. Although the literature indicates that ultrasound exhibits high specificity in diagnosing tendon ruptures, as tendon discontinuity and retraction of the torn ends (*Habib et al., 2023*), our findings reveal contextual limitations in complex wrist anatomy. The intricate anatomy of the wrist, particularly in the presence of joint effusion and tissue swelling, poses challenges in the accurate identification of ruptures. Our study revealed that without passive motion during the examination the diagnostic accuracy of ultrasound for ET and FT ruptures was 50.00% (9/18) and 21.43% (3/14), respectively. However, after passive motion diagnostic accuracy significantly improved to 92.86% (13/18) and 100% (14/14) for ET and FT ruptures, respectively. The difficulty in visualizing the severed end of a ruptured ET is attributed to several factors, such as some patients receive immobilization treatment prior to early-stage ultrasound examination. This immobilization aligns the tendon in an anatomical position, complicating its visualization. Additionally, the ET is relatively thin, and the surrounding structures provide limited space, further obscuring the severed end. Passive motion of the finger's distal can enlarges the space around the tendon, enhancing the exposure of the ET's end and improving the accuracy of ultrasound diagnosis (*Chen et al., 2023*).

Distinguishing between FDP and FDS is crucial as they have distinct functions at the finger level. Assessing FT injuries not only involves determining the presence of rupture but also aids in identifying the level and extent of the injury, guiding surgical decision-making. Among the 16 patients with FT injuries, 37.50% (6/16) exhibited normal or minor tears in the FDS combined ruptures in the FDP. Passive motion of the finger proved valuable in discerning the extent of tendon damage, enhancing the accuracy of imaging diagnosis for clinical management (*Mori & Teng, 2023*).

The diagnostic accuracy for ET tears was lower than ET ruptures, at 16.67% (1/16) before and 33.33% (2/16) after passive motion, respectively. This may be due to the ET is thinness, which complicates the ultrasound detection of subtle muscle fiber changes within the thinness tendon. Additionally, the persistence of passive motion in the injured tendon despite occurred tear make it further complicates in ultrasound diagnosis. Among the six patients with ET tears in the study, only two were correctly diagnosed by ultrasound. Among the four misdiagnosed patients, one was found to have an ET tear during surgery, despite an ultrasound indicating a rupture. This discrepancy may be attributed to a hematoma surrounding the tendon rupture, obscuring parts of the tendon that remained intact, and causing a false-negative in passive motion tests. In the other three patients diagnosed with ET tears, ultrasound images revealed swollen tendons with well-done passive motion. This could be due to small blood vessel congestion from the acute injury, which obscured minor muscle fiber tears within the swollen tendons (*Haidar et al., 2023*). These findings suggest

that patients with tendon swelling should undergo high-resolution dynamic ultrasound to assess for subtle fiber tears, including evaluation for fiber discontinuity during passive motion and hypervascularity on Doppler. Critical tears (*e.g.*, >50% thickness or gap >5 mm) may require surgical repair, while minor tears could be managed conservatively with splinting and early mobilization. Furthermore, the study revealed a modest ultrasound accuracy of 40.00% (2/5) in diagnosing nerve injuries. The median nerve injury was clearly diagnosed in this study, while only one out of four cases of digital nerve injury was identified. It may be related to the relative thinness of the digital nerve and the lack of diagnostic specificity of the ultrasound images when swelling has not occurred. Furthermore, there is few of ultrasound research focusing on digital nerve injuries, which has contributed to the limited diagnostic experience among sonographers in identifying digital nerve morphological abnormality (*Wu et al., 2023*). Future clinical examinations should emphasize more meticulous assessment of digital nerves to enhance diagnostic sensitivity for digital nerve disorders.

Four patients in the study presented with avulsion fractures, all of which occurred in the ET at the attachment site of the DP, a commonly reported location for avulsion fractures in traumatic hand patients (*Choke, Tan & Cheah, 2023*). Avulsion fractures are identifiable on ultrasound by the presence of cortex irregularity at the tendon injury, along with freely bone fragments in the surrounding area. Studies have shown that ultrasound exhibits superior diagnostic sensitivity for avulsion fractures compared to X-ray imaging and is generally consistent with the sensitivity of three-dimensional CT reconstruction (*Li et al., 2023*; *Crombach et al., 2020*). Therefore, when dealing with traumatic hand patients, a detailed observation of the bone cortex at the tendon in ultrasound is essential to detect subtle avulsion fractures that may undetected on X-ray.

## LIMITATIONS

This study has several limitations. Firstly, the sample size of 47 cases may restrict the generalizability of the descriptive findings, and future studies with larger cohorts are needed. Secondly, the variations in ultrasound images at different injury time points were not explored. Thirdly, the study did not stratify analyses by injury duration, which may influence ultrasound findings and diagnostic accuracy. This limitation restricts the ability to draw conclusions about time-dependent variations. Finally, the absence of continuous, standardized postoperative patient follow-up to analysis whether ultrasound can evaluate the prognosis in hand trauma. We will continue to focus on these in the future research.

## CONCLUSION

In summary, high-frequency musculoskeletal ultrasound is highly valuable for diagnosing tendon ruptures and avulsion fractures in hand trauma, added passive motion can enhance the diagnostic accuracy for tendon ruptures during examination. Ultrasound can provide

reliable imaging evidence for clinical clarification of the type and extent of injury, to guide adjunctive treatment selection for patients with hand trauma.

### Funding
This work was supported by Anhui Province clinical medical research transformation project (No. 202304295107020016). The funders had no role in study design, data collection and analysis, decision to publish, or preparation of the manuscript.

### Grant Disclosures
The following grant information was disclosed by the authors:
Anhui Province clinical medical research transformation project: No. 202304295107020016.

### Competing Interests
The authors declare there are no competing interests.

### Author Contributions
- Jiajia Wang conceived and designed the experiments, performed the experiments, analyzed the data, prepared figures and/or tables, authored or reviewed drafts of the article, and approved the final draft.
- Lan Gao conceived and designed the experiments, performed the experiments, analyzed the data, prepared figures and/or tables, authored or reviewed drafts of the article, and approved the final draft.
- Mingdi Fang analyzed the data, prepared figures and/or tables, and approved the final draft.
- Fan Jiang analyzed the data, prepared figures and/or tables, and approved the final draft.
- Mei Peng conceived and designed the experiments, authored or reviewed drafts of the article, and approved the final draft.

### Human Ethics
The following information was supplied relating to ethical approvals (i.e., approving body and any reference numbers):
The Second Affiliated Hospital of Anhui Medical University granted ethical approval to carry out the study within its facilities.(Ethical Application Ref: SL-YX2024-052)

### Data Availability
The raw measurements are available in the Supplementary Files.

### Supplemental Information
Supplemental information for this article can be found online at http://dx.doi.org/10.7717/peerj.20514#supplemental-information.

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
