# Peer review of "Accuracy of high-frequency musculoskeletal ultrasound in hand trauma: a retrospective surgical field-based validation study at a single center"

_PeerJ, doi:10.7717/peerj.20514_

## Round 0.1 · original submission · Major Revisions

· Academic Editor

Major Revisions

**Language Note:** The review process has identified that the English language must be improved. PeerJ can provide language editing services - please contact us at [email protected] for pricing (be sure to provide your manuscript number and title). Alternatively, you should make your own arrangements to improve the language quality and provide details in your response letter. – PeerJ Staff

Reviewer 1 ·

Basic reporting

This is a single-center, retrospective study of the diagnostic accuracy of musculoskeletal ultrasound in hand trauma. A retrospective analysis of 47 cases of hand trauma were analyzed which have both ultrasound imaging diagnosis and surgery record. The authors found that ultrasound can diagnose tendon rupture and finger avulsion fracture with high accuracy. However,due to the different injury times and symptoms of hand injuries, the purposes of ultrasound examinations are not the same. Simply reporting the accuracy of ultrasound in diagnosing tendon ruptures and avulsion fractures has no practical clinical value and is of little reference significance to peers.

Experimental design

The following comments should be mentioned:
1. The introduction of the research background is insufficient, and the reference is selected and cited not properly.
IN INTRODUCTION SECTION, Line1-3 “Hand trauma is a frequent sports-related injury…..(De 37 Jong et al., 2014).” This expression is incorrect. Hand trauma is not a common sports injury, and this expression cannot be found in the cited literature either.
Line42-43 “However, this method is inherently subjective, may not accurately determine the precise location and the extent of tendon injuries, impeding surgical treatment (Koehne et al., 2023)” This sentence also has no relevant content in the cited literature。
Line44-46 “While X-ray and CT are useful for detecting hand fractures, they have limited sensitivity for avulsion fractures and low detection rates for tendon and nerve injuries in the hand (Von Schneider Egestorf et al., 2017)” No relevant content can be found in the cited literature and the statement is not correct. In fact, CT has high sensitivity and specificity in diagnosing avulsion fractures.
2. As a study for evaluating ultrasound images, there is no description of the specific ultrasound signs and corresponding definitions for tendon rupture, tendon tear, and tendon swell. May I ask what specific pathological condition tendon tear represents and what the difference is from partial rupture? What do "tendon thin" and "tendon swell" represent? How is surgical field-based validation (see title of the manuscript) specifically defined? Shouldn't it be an intraoperative or postoperative diagnosis?
3. The English writing language of the manuscript contains inaccurate word usage and unclear expression, and some parts are difficult to understand. The English language should be improved to ensure that an international audience can clearly understand the text.

Validity of the findings

no comment

Additional comments

no comment

Reviewer 2 ·

Basic reporting

The aims of study was to assess the diagnostic accuracy of high-frequency musculoskeletal ultrasound in hand trauma patients, with a specific focus on tendon injury, avulsion fracture and nerve damage, compared by surgical observations. The study is study is well structured, with adequate references.
However, the discussion must be in relation to its results.

Experimental design

with respect to the experimental design, the n of the groups is unbalanced. In acordding with results, the analyis can be descriptive.

Validity of the findings

The results are valid. Please review the way the data is described.

Additional comments

The paper is well-written. Please review the English language, the detailed description of your results, and ensure the discussion is guided by your findings.

·

Basic reporting

The writing of the method section changes (line 81) to read like a short hand protocol in present tense. the remainder of the method is in past tense.

line 146 please explain radial carpal flexor

line 218 this sentence appears incomplete

the authors have reported % but also raw numbers which is valuable, as subgroup numbers are low.

line 244 write numbers below 10 as a word eg six, not a digit 6

line 252 can you please elaborate on how to assess for potential localized muscle fiber tears and how would this alter your treatment plan?

line 260 the sentence appears incomplete

Experimental design

This paper could be enhanced by comparing ultrasound findings with clinical diagnostic findings that were made pre-surgery. It's understood that the surgeries confirmed site and structures injured plus proportion of the tendon injured, as compared with ultrasound but what about the initial examination - how did this compare with the surgical findings re location of injury..

did the ultrasound change the surgical field that was planned from the clinical exam or any other aspect of patient care.

Validity of the findings

The authors are numerically honest and it was appropriate to apply only basic descriptive statistics to these findings

Additional comments

This interesting paper could be enhanced with further explanation and comparison by incorporating clinical findings into the analysis.

---

## Round 0.2 · Minor Revisions

· Academic Editor

Minor Revisions

Thank you for revising your manuscript in response to the concerns raised by our reviewers. Although reviewer 2 now recommends publication, reviewer 1 remains concerned about the clinical relevance of your study, amongst other things. We therefore ask that you submit a further revision of the manuscript that addresses the various points raised by this reviewer, which we envisage will include additional text to explain the limitations of your work with respect to clinical practice, in addition to other appropriate revisions.

Reviewer 1 ·

Basic reporting

The revised manuscript entitled “Accuracy of high-frequency musculoskeletal ultrasound in hand trauma: A retrospective surgical field-based validation study: A single center experience” tried to report a single-center, retrospective observational study of the diagnostic accuracy of musculoskeletal ultrasound in hand trauma. A retrospective analysis of 47 cases of hand trauma were analyzed which have both ultrasound imaging diagnosis and surgery record. The authors found that ultrasound can diagnose tendon rupture and finger avulsion fracture with high accuracy.

Experimental design

However,Hand trauma is commonly encountered in the emergency department. Tendon rupture of hand requires emergency surgical intervention to avoid possible disability and dysfunction and restore hand function as much as possible. For patients with acute hand tendon rupture, clinical history and physical examination can provide sufficient clues for correct diagnosis. The purpose or advantages of ultrasound examination is to determine the location of the ruptured tendon and the degree of separation of the ruptured ends, which is conducive to the determination and implementation of the surgical plan. For patients in the chronic stage of hand tendon rupture, due to local scarring and tissue adhesion, the possibility of tendon suture repair is very small. Tendon implantation may be an effective method. The value of ultrasound also lies in evaluating the location of the rupture ends, the separation situation, and the local scar adhesion situation, helping to formulate a surgical plan with necessary information. Therefore, due to the different injury times and symptoms of hand injuries, the purposes of ultrasound examinations are not the same. Simply reporting the accuracy of ultrasound in diagnosing tendon ruptures and avulsion fractures has no practical clinical value and is of little reference significance to peers. Not to mention that the number of cases was relatively small (only 47 cases), and there was no distinction made between the duration of patients' injuries.

Validity of the findings

no comment

Additional comments

1. The comments I have mentioned before were not revised correctly or appropriately. There still had several referents selected and cited improperly. (like:Line105)
2. As a study for evaluating ultrasound images, the description of ultrasound signs and corresponding definitions were not in accordance with clinical guidelines.

Reviewer 2 ·

Basic reporting

no comment

Experimental design

no comment

Validity of the findings

no comment

Additional comments

The authors have improved the article

---

## Round 0.3 · Minor Revisions

· Academic Editor

Minor Revisions

Thank you for submitting a further revision of your manuscript and a response to reviewer 1's comments from the previous round of review. Although revisions have been made to the manuscript to address some of the concerns, we note that others remain unaddressed. We therefore require submission of a further revision of your manuscript that includes appropriate changes to address all the reviewers' concerns. The resubmission should be accompanied by a point-by-point response that responds to the various points raised and also explains how you have revised the manuscript to address these concerns.

---

## Round 0.4 · Minor Revisions

· Academic Editor

Minor Revisions

Thank you for the further revision of your manuscript in response to reviewer 1's comments from the earlier round of review. Although we appreciate that you have modified the references cited, we note that no other changes have been made to the manuscript in response to this reviewer's comments. For example, it appears that no changes have been made to tone down claims of statistical significance to focus on descriptive outcomes, as you suggest in your response document. We therefore require submission of a further revision of your manuscript that includes appropriate changes to address all the reviewer's concerns.

The resubmission should be accompanied by a point-by-point response that responds to the various points raised, accurately explaining how the resubmitted manuscript has been revised. Please note, however, we would be unlikely to offer you more than this one last chance to adequately address this reviewer's concerns in your revised manuscript.

---

## Round 0.5 · accepted · Accept

· Academic Editor

Accept

Thank you for revising your manuscript in response to reviewer 1. I am now satisfied that the reviewers' comments have been satisfactorily addressed. The manuscript is now ready for publication.